# Critical Illness in Migrant Workers in the Windsor-Essex Region: A Descriptive Analysis

**DOI:** 10.3390/ijerph20166587

**Published:** 2023-08-16

**Authors:** Alex Zhou, Abdelhady Osman, Genesis Flores, Dhuvaraha Srikrishnaraj, Jayashree Mohanty, Retage Al Bader, Amy Llancari, Aya El-Hashemi, Manahel Elias, Kanza Mirza, Maureen Muldoon, Ryan Palazzolo, Farwa Zaib, Indryas Woldie, Caroline Hamm

**Affiliations:** 1Schulich School of Medicine, Dentistry Windsor Campus, Windsor, ON N9B 3P4, Canada; azhou2025@meds.uwo.ca (A.Z.); aosman2025@meds.uwo.ca (A.O.); dsrikrishnaraj2025@meds.uwo.ca (D.S.); aalbader2025@meds.uwo.ca (R.A.B.); melias2025@meds.uwo.ca (M.E.); kmirza2025@meds.uwo.ca (K.M.); fzaib2025@meds.uwo.ca (F.Z.); 2Department of Biomedical Sciences, University of Windsor, Windsor, ON N9B 3P4, Canada; flores51@uwindsor.ca (G.F.); aya@uwindsor.ca (A.E.-H.); palazzor@uwindsor.ca (R.P.); indryas.woldie@wrh.on.ca (I.W.); 3School of Social Work, University of Windsor, Windsor, ON N9A 0C5, Canada; jayashree.mohanty@uwindsor.ca; 4Department of Biological Sciences, University of Windsor, Windsor, ON N9B 3P4, Canada; llancara@uwindsor.ca; 5Faculty of Arts, Humanities and Social Sciences, University of Windsor, Windsor, ON N9B 3P4, Canada; mhmul@uwindsor.ca; 6Windsor Regional Hospital Cancer Program, Windsor, ON N8W 1L9, Canada

**Keywords:** critical illness, migrant workers, medical repatriation, cancer, Windsor-Essex

## Abstract

Despite their essential role in Canadian agriculture, migrant workers face numerous healthcare barriers. There is a knowledge gap regarding the healthcare experiences of migrant workers with critical illness in the Windsor-Essex region. Our objective was to collect information on the experiences of migrant workers experiencing a critical illness at Windsor Regional Hospital (WRH) between 31 December 2011 and 31 December 2021. We conducted a retrospective chart review and interviews with migrant workers. We identified 14 migrant workers who presented to WRH with a critical illness over these 10 years. Despite occasional barriers regarding access to care, the migrant workers received an appropriate standard of care in Canada. Five of the fourteen patients identified were repatriated to their home countries. The migrant worker patients interviewed expressed satisfaction with the care they received in Canada but identified repatriation as a specific concern to receiving continuity of care. The health and financial burden imposed by critical illness on migrant workers and their employers makes critically ill workers vulnerable to medical repatriation as a unique social determinant of health. Considering the critical role of migrant workers in Canada’s food security, policy changes should be considered to ensure critically ill workers are able to remain until recovery.

## 1. Introduction

Migrant farm workers are essential to Canada’s food security. In 2020, 50,126 foreign workers were hired to work in Canada’s agricultural sector [1]. They were hired primarily through the Seasonal Agricultural Worker Program (SAWP) and the Agricultural Stream of the Pilot Project for Occupations Requiring Lower Levels of Formal Training [2]. The Windsor-Essex region hosts one of the largest communities of migrant farm workers in Canada, with 8000–10,000 hired to work across the region [3]. Migrant workers face numerous barriers to healthcare access, including language and cultural barriers, poor health literacy, insufficient time to seek care, and reliance on overstressed rural healthcare facilities [4,5,6,7]. While workers are eligible for provincial health insurance, they depend on their employers for access to care. Employers often control the workers’ access to their health documents, work hours, transportation, and communication with healthcare providers [4,5,6,7]. Employers are also required to provide workers with coverage for non-insured services and coverage during mandatory waiting periods before they become eligible for provincial health insurance. Migrant workers may choose not to seek care, out of fear that their health status could result in termination and/or repatriation back to their home country [7]

Previous studies show that migrant workers in Canada have been previously repatriated for becoming ill, injured, or pregnant [5,7,8]. More specifically, medical repatriation in Ontario was studied using data from Foreign Agricultural Resource Management Services, a non-profit corporation managing the contracts of more than 15,000 migrant farm workers in Ontario annually. Migrant farm workers were most frequently repatriated for medical or surgical reasons (41.3%, n = 325), for external injuries including poisoning (25.5%, n = 201), and for other identifiable reasons (17.3%, n = 136). Of the 787 medical repatriations in this time period, 5.8%, or 45 workers over 10 years, were listed as cancer and maybe considered life-threatening. The other medical reasons would likely be interpreted as non-life-threatening [8]. More recently, a survey of migrant workers’ health in British Columbia found similar findings to those in Ontario. Mainly, migrant workers complained of communication barriers, a lack of confidence in the healthcare system, a lack of contact with advocacy or support systems, and fear of medical repatriation [9]. After repatriation, critically ill workers may not be able to continue treatment in their home country for financial reasons or ineligibility [10].

While most of the previous research has focused on migrant worker health in general, there is a more specific knowledge gap regarding the experiences of migrant workers who develop critical or life-threatening illnesses during their employment in Canada, as they tend to be most vulnerable to medical repatriation [4,6]. In addition, there is a knowledge gap regarding the experiences of critically ill migrant workers in the Windsor-Essex region, which is home to one of the largest communities of migrant workers in Canada.

The purpose of this descriptive study was to present qualitative data describing the experiences of migrant workers who were treated for critical illnesses at the referral community hospital in the Windsor-Essex region between 2011 and 2021 through a retrospective chart review and patient interviews.

## 2. Materials and Methods

The data were collected through a combination of a retrospective chart review and patient interviews. The retrospective chart review was completed by the authors A.Z. and A.O., while the patient interviews were conducted by D.S. and G.F. The methodology for both processes is described below. The study was approved by the WRH Research Ethics Board REB numbers #22-427 and 22-441.

### 2.1. Patient Identification and Eligibility

For this retrospective chart review, critical illness was defined as a “life-threatening illness or trauma”. Employers hiring migrant farm workers through SAWP or the Agricultural Stream are required by the federal government to provide onsite housing for workers. Workers are legally required to reside in this employer-provided onsite housing [11]. To identify migrant worker patients, a list of farm and greenhouse addresses in Windsor, LaSalle, Essex, Kingsville, and Leamington was created based on an online search. Patient charts listing those addresses were retrieved from electronic medical records at Windsor Regional Hospital (WRH). From these charts, patients were confirmed as migrant workers through the social history portion of their medical charts, with only charts explicitly identifying the patient as a migrant or seasonal worker originating from a foreign country included. The patient charts were reviewed by the authors A.Z. and A.O. to determine if they met the criteria of having a critical illness, as previously defined. The charts were only reviewed further if they explicitly identified the patient as a migrant worker. Any chart that did not explicitly identify the patient as a migrant worker was excluded from further analysis. Additionally, charts belonging to a cohort of critically ill cancer patients identified as migrant workers were contributed for inclusion in this study by the authors I.W. and C.H.

Migrant worker patients who were undergoing treatment for a critical illness and were willing to be interviewed about their experiences were identified by I.W. and C.H. and contacted by D.S. and G.F. Consent forms were reviewed and submitted by the patients prior to the interviews, with the option to withdraw from study until the conclusion of the interview

### 2.2. Data Collection

For the retrospective chart review, the data from patient charts were collected and analyzed using REDCap data capture tools hosted at the University of Windsor. Prior to collection, the patient data were de-identified, except for the patient MRN and date of birth (DOB), which were retained as identifier fields. The following variables were extracted from each chart: address, age, sex, country of origin, employer/type of work, initial point of contact with the healthcare system, health insurance type, patient social history, and patient disposition. Any missing information deemed relevant to the purpose of the study was recorded under a “knowledge gap” variable. All the variables were entered as free text or multiple-choice options in REDCap.

Eligible patients who consented to be interviewed were interviewed over the phone by D.S. and G.F. in English and Spanish. The patients were asked a series of standardized questions. Specifically, the migrant worker patients were asked about the timeline of illness diagnoses, treatment experience, social support they received, and challenges they faced. The full list of questions has been included in Appendix A. The responses were recorded, transcribed, and translated into English by D.S. and G.F.

The data collected from the retrospective chart review and patient interviews were presented using a qualitative method. For the retrospective chart review, all the charts were reviewed independently by A.Z. and A.O. to identify common themes. Emphasis was placed on the patient’s social history, health insurance type, initial point of contact with the healthcare system, and patient disposition. The patient interview transcripts were initially reviewed by D.S. and G.F. to identify key themes regarding the patient’s experiences with critical illness. The transcripts were further reviewed by A.Z. and A.O. to integrate these findings with the themes identified in the retrospective chart review. Any disagreements were resolved by discussion with I.W. and C.H.

## 3. Results

Sixty patient charts were reviewed in this study. Fifty-four charts were retrieved based on the list of greenhouses in the Windsor-Essex area compiled by the authors. Additionally, six charts were contributed to this study by the authors I.W. and C.H. Of the 60 charts reviewed, we identified 14 (16%) migrant worker patients with critical illnesses meeting our criteria, who presented to the local hospital between 31 December 2011 and 31 December 2021. The full chart selection results are displayed in Figure 1.

The patients had a median age of 41 years, and all were male. All the patients presented to the emergency department (ED) as their initial point of contact with the healthcare system. The most common country of origin of the patients was Mexico (n = 6). Five of the fourteen patients identified were repatriated to their home countries during or after treatment. The patient demographic characteristics and diagnoses are displayed in Table 1.

Through a review of all the patients’ charts, we identified two main themes related to the patients’ healthcare experience at WRH. These include care while in Canada and medical repatriation before recovery.

### 3.1. Care While in Canada

All 14 patients included presented to the ED as their initial point of contact with the healthcare system. None had a family physician recorded on their charts. One case involved a 46-year-old migrant worker presenting to the ED feeling unwell and dizzy. Following investigations, he was diagnosed with acute myeloid leukemia. Electronic Medical Record (EMR) documents indicate he was told there was a high risk of death within 30 days. No further documents on the medical history indicating the outcome for this patient were found.

All 14 patients included in this analysis were initially insured under Ontario Health Insurance Plan (OHIP) but often did not maintain OHIP after diagnosis for the duration of their treatment. Migrant workers do appear to face challenges with health insurance when their Canadian work permit expires or when they leave their initial employer. We report a 32-year-old migrant worker in our search who was diagnosed with leukemia and appealed to remain in Canada to continue treatment after he left his initial employer and his work permit had expired. He is currently appealing to remain in Canada under compassionate grounds. The patient lost his OHIP eligibility after his work permit expired and received his health care only through the provincial Enhanced Health Care Coverage Critical to Support Efforts to Contain COVID-19, which was rolled out in 2020 and has since been suspended in 2023. This worker was aware that his prognosis would be only 2–4 years if he were denied ongoing treatment in Canada since treatment would not be available to the patient in Jamaica because of financial constraints. In contrast, remaining in Canada would allow treatment with the goal to cure his leukemia.

Seven of the fourteen migrant worker patients included in this analysis spoke little or no English according to the medical records. The documentation in the patient charts indicated that translators were provided for the migrant workers that could not speak English. The exception was the case of a 47-year-old migrant worker who was diagnosed with a subarachnoid hemorrhage; the ER physician noted that the initial exam was limited due to a language barrier. The patient received surgery and fully recovered but did not show up for follow-up appointments. The status of the patient’s further stay in Canada is unknown at the time of writing.

As part of this study, two migrant worker patients were interviewed about the care they received while in Canada. Notably, both patients expressed appreciation for the support received from employers and healthcare workers for medical treatment and logistical support.

“Employer was supportive of me. He gave me lighter duties when he found out I was sick and gave me transport to the hospital … my employer did everything that he could’ve done”.(32-year-old male migrant worker patient, undergoing treatment for leukemia)

“They [the social worker] provided transportation whenever I needed. They shared [transportation duties] with my boss. … They helped me with a program that allowed me to still get the medications … [My oncologist] was very supportive … The nurses were also very helpful”.(32-year-old male migrant worker patient, undergoing treatment for leukemia)

“After that visit in Emergency, my health has deteriorated. I thank [employer’s name], the foreman and my employer for being supportive during that time … I was moved to another bunkhouse close to the farm where my coworkers could keep an eye on me and help me when necessary”.(61-year-old male migrant worker patient who has been coming to Canada for 18 years, undergoing treatment for prostate cancer)

### 3.2. Medical Repatriation of Migrant Worker Patients Before Recovery

The 14 migrant worker patients with critical illnesses met various healthcare outcomes. The outcomes of their treatments varied as some patients received care to full recovery, while others were subject to transfer to other treatment centers or repatriation to their home countries, in which follow-up to care was lost and the final patient disposition could not be determined. As of the end of the chart review period (7 July 2022), two of the fourteen migrant worker patient cases examined recovered and returned to work following treatment at WRH. Two patients recovered but were lost to follow-up following recovery. One patient received corrective surgery with no evidence of sequelae but was lost to post-operative follow-up. Two of the fourteen patients were still receiving ongoing care for their critical illnesses as of the completion of this chart review. One additional patient was notable in that they received palliative care in Canada for their aggressive condition, with the Mexican consulate providing support by transporting his family to visit him prior to his death. Five of the fourteen migrant worker patients who were diagnosed with critical illnesses were repatriated to their countries of origin, with the expectation to follow up with healthcare systems there. Specifically, they were diagnosed with leukemias, lymphomas, metastatic germ cell tumors, and suspected Miller Fisher syndrome. Critical illness also imposes significant strain on the employers of migrant workers. In one case, a 31-year-old migrant worker was diagnosed with a metastatic germ cell tumor. The patient’s Canadian work permit was due to expire, and he was scheduled to return to Mexico. The patient’s employer provided private accommodation on the farm pending the patient’s return. However, that meant that the rest of the workers had to be moved to suboptimal housing. As of the conclusion of the chart review, the patient’s repatriation was scheduled for September 2022.

Of the patients who were repatriated, all the patient charts reviewed received a working diagnosis before repatriation, with one notable exception. One case described a 48-year-old migrant worker who presented with ptosis and ophthalmoplegia and was suspected of having a stroke. The patient’s work term in Canada was near completion and he was due to return to Mexico before a formal diagnosis could be made. A stroke was ruled out, and the patient was determined to be medically stable. The patient was suspected to have Miller Fisher syndrome, a rare neurological condition causing paralysis of the eye muscles and a lack of reflexes. He was discharged and advised to follow up with a local neurologist in Mexico for a definitive diagnosis. The patient was provided with copies of all the diagnostic workup that was done at WRH in paper format, although no follow-up had been arranged with a neurologist in Mexico given the short timeframe.

Of the migrant worker patients reviewed in the chart, two of them were interviewed as of the time of writing (10 Mar 2023). During the interviews, the migrant workers expressed significant distress over potential healthcare coverage loss and possible repatriation to their countries of origin. Specifically, one worker expressed concern that they would not receive a similar standard of care if they returned.

“Also, if I had gone back, I wouldn’t get this treatment I’m getting now. And if they exit me off the program, I wouldn’t be able to come back either [to Canada]”.(32-year-old male migrant worker patient, undergoing treatment for acute leukemia)

Additionally, the patient expressed concern over the impact of the critical illness on their work status in Canada. This included some confusion on the patient’s part regarding how healthcare coverage was provided for them and access to healthcare.

“…if I hadn’t applied for open work permit, then I’d be worried because I would be here illegally. That was the only reason I was worried, of ending up being here illegally. The social workers and the doctors helped me with open work permit application”.(32-year-old male migrant worker patient, undergoing treatment for leukemia)

“In the future, honestly, what was scaring me was that my permit would have been expired, I wouldn’t be able to get health coverage”.(32-year-old male migrant worker patient, undergoing treatment for leukemia)

Moreover, both patients interviewed expressed a preference to remain in Canada for treatment.

“In the future, if someone’s permit expires on them, I would like them to get the health support and coverage. I would like if they wouldn’t have to be forced to go back home. I would like if they can get the health support here, no matter what their status is”.(32-year-old male migrant worker patient, undergoing treatment for leukemia)

“I am requesting that the Canadian government extend my temporary resident permit until I have had a couple of injections from my treatment plan in order to help stabilize my health”.(61-year-old male migrant worker patient who has been coming to Canada for 18 years, undergoing treatment for prostate cancer)

## 4. Discussion

We identified major themes in migrant worker healthcare provision. All 14 patients identified presented to the ED with acute symptoms. In the patient histories, there was no record of the migrant workers having a family physician, indicating a possible lack of access to primary care for migrant workers. Their eligibility for provincial health insurance plans (such as OHIP) is often dependent on their employer, and those unable to work may lose coverage or avoid care due to a fear of repatriation back to their home countries [4]. Moreover, our chart review noted several patients who required interpretation services or whose exams and understanding of the investigation results were limited by language barriers. Language barriers may compromise patients’ ability to navigate the Canadian healthcare system and physician–patient communication, impeding the willingness of migrant worker patients to seek care due to a perceived lack of cultural safety [12]. A combination of these barriers may play a role in dissuading migrant workers from seeking care until the symptoms are acute and severely distressing, culminating in ED visits rather than primary care as a first measure.

Through this review, we noted incomplete or conflicting social histories documented for some migrant worker patients. As an example, one patient had OHIP listed on demographic information as health coverage. However, a consultation note revealed this patient had no health coverage. It is unknown whether this patient had received health coverage and lost it because of losing their work status or if this was improperly documented during the initial intake. Under Canadian federal law, migrant workers are eligible for provincial health insurance [13]. However, this access may be delayed and is directly linked to the employment status and is dependent on the employers [6,13]. Moreover, this patient was noted to be in an unstable common-law relationship, with no further indication of whether social services aid was provided. Insufficient social histories in patient charts may present a limitation regarding understanding the impact of disease on the work status, insurance coverage, and overall individual illness experience of the patients.

As suggested through the chart review, a significant proportion of migrant worker patients were repatriated to their countries of origin. Additionally, as observed in the interviews, the possibility of medical repatriation is a consistent concern of migrant worker patients. Fear of deportation is a well-established proximal determinant of health in the health of migrant workers. Specifically, it is known that fear of repatriation is a contributing factor to migrant workers not seeking care or is detrimental to their mental health [14,15]. Furthermore, loss of the ability to work often jeopardizes the healthcare coverage of migrant patient workers and can lead to the inability to claim compensation following their return to their countries of origin [5]. A significant portion of the migrant worker patients reviewed returned to their countries of origin. Repatriation results in an inability to maintain follow-up with the patient and discontinuity of care, specifically in Canadian standards of care. This presents some complicated ethical dilemmas, especially regarding the impact on their care and the autonomy of migrant workers’ preferences in making their healthcare decisions regarding location, quality, and provision. Barriers to accessing healthcare, such as language barriers and a lack of primary care, can exacerbate vulnerability to medical repatriation, as migrant workers lack access to social, legal, and advocacy opportunities provided by the healthcare system. As shown in the interviews conducted in this study, the migrant worker patients expressed appreciation for the social and legal support provided by members of the healthcare team during their treatment in Canada. This included assistance in maintaining healthcare coverage and maintaining legal status in Canada to avoid or defer repatriation.

Critical illnesses impose costs on the employers of migrant workers. As shown in the interviews and the chart review conducted in the study, the employers provided special accommodation and transport for the patients during treatment. The provision of long-term care support for critically ill migrant workers by employers is generally unsustainable, and this increases the vulnerability of migrant workers to medical repatriation.

### 4.1. Strengths and limitations

To the best of our knowledge, this is the first study on the experiences of migrant workers with critical illness in Windsor-Essex. The study includes a descriptive analysis of the experiences of critically ill migrant workers using data from their electronic medical records as well as a description of their experiences in their own words through interviews.

However, our study is not free of challenges. Our study is limited by a relatively small sample size. This is due to several reasons. There are no unique identifiers that could be used to retrieve the charts of all the migrant workers who were seen at WRH between 2011 and 2021. As a result, we had to retrieve the charts using patient addresses. We conducted a search of all the greenhouses and farms whose address was publicly listed and retrieved the charts of all the patients who presented to WRH from those addresses between 2011 and 2021. Additionally, we used narrow inclusion criteria for the patient charts to be included in qualitative analysis. Patient charts were excluded if there was no explicit documentation in their social histories showing they were a migrant worker. As a result, our review may not be exhaustive of all the critically ill migrant workers who were seen at WRH during the specified time period.

A limitation of this retrospective chart review analysis is that information on several pertinent social history detail factors was missed. This includes information on the patients’ immigration status in Canada, work conditions, relationships with their employers in Canada, financial situations, or social support systems in Canada.

Another limitation of our study is the small number of interviews. At the time of writing, only two migrant worker patients had been recruited to be interviewed about their experiences with critical illness. While data saturation cannot be achieved with two interviews, we judged that it would be important to include a description of the migrant workers’ experiences in their own words, which could not be achieved with the chart review only. Recruitment for more migrant workers to be interviewed is still ongoing.

All the critically ill migrant worker patients who were considered in this study were male. This is consistent with previous findings that show that the majority of migrant workers in Canada are male [8]. Nonetheless, the lack of representation of the experiences of critically ill female migrant workers is a limitation of the study.

Moreover, our study should be interpreted strictly as a description of critical illness in migrant workers. It does not include analyses of migrant workers presenting with occupational hazards or musculoskeletal issues, which account for the majority of migrant worker clinical experiences [8]. Additionally, while the legal, social, and ethical aspects of medical repatriation are relevant, they cannot sufficiently be addressed by this study.

### 4.2. Future Directions

Future directions for this review include expanding its scope to include other local hospitals in the region, particularly ErieShores Healthcare in Leamington, as it hosts a large community of migrant workers. We also aim to increase the number of interviews conducted with eligible migrant worker patients about their experiences with critical illness in Canada, as patient recruitment is ongoing.

The findings of this review will be used to create a survey for migrant workers examining their attitudes towards their healthcare experiences and the barriers they face to receiving care and a survey on the moral distress experienced by healthcare workers treating migrant workers. The findings of this review will also be presented to engage and partner with local, provincial, and federal stakeholders and decision-makers on initiatives to improve the healthcare experience of migrant workers, the distribution of surveys, and preliminary planning to provide primary care access for migrant workers. Ultimately, the aim of these partnerships is to advocate for policy changes to provide financial support and healthcare access to critically ill migrant workers and their employers, which would allow them to continue treatment in Canada in a manner that is consistent with the Canadian principles of universality and accessibility. The population of migrant workers who are medically repatriated for critical illnesses, such as malignancies, is small, as shown in this review and previous studies [7]. In addition, migrant workers represent a structural necessity in Canadian food security. The Canadian Charter of Rights and Freedoms guarantees the right to life, liberty, and security of persons residing in Canada. Moreover, Canada is a signatory to international agreements that guarantee the safety and security of migrant workers. This includes being a signatory to international agreements that guarantee the safety and security of migrant workers at the Universal Declaration of Human Rights (UDHR) and the UN International Covenant on Economic, Social, and Cultural Rights (ICESCR) [16].

The critically ill migrant workers interviewed have expressed a desire to remain in Canada for the duration of their treatment. Our working group feels it would not represent an overwhelming financial burden to ensure continued healthcare coverage and support for this group of migrant workers considering the small numbers identified in this study. This would ensure the continuation of the standards of care and prevent loss of follow-up in the care of these patients.

## 5. Conclusions

This is the first study, to our knowledge, that focuses on the healthcare experiences of critically ill migrant workers in the Windsor-Essex region and Canada. This is especially significant since Windsor-Essex is host to a sizeable community of migrant workers yearly and contains one of the largest communities nationally. Our chart review and interviews found that critically ill migrant workers generally receive medical care congruent with the Canadian standard of care while in Canada. The perception of the interviewed migrant workers of their own healthcare experience while in Canada concurs with this assessment. Our study highlights medical repatriation as a persistent concern for critically ill migrant workers in the Windsor-Essex region. This supports the existing knowledge of repatriation as an often unique determinant of health for migrant workers across Canada. This concern was reflected in the interviews. Public health policy changes should focus on allowing critically ill migrant workers to receive the unencumbered option to remain in Canada for healthcare if they become ill during their work term in Canada. Specifically, the findings of this study highlight broader issues around the continuation of healthcare coverage for vulnerable populations. Until public health policy changes are enacted, medical repatriation and a lack of coverage remain obstacles to the Canadian healthcare principles of universality and accessibility while impinging on patient autonomy.

## Figures and Tables

**Figure 1 ijerph-20-06587-f001:**
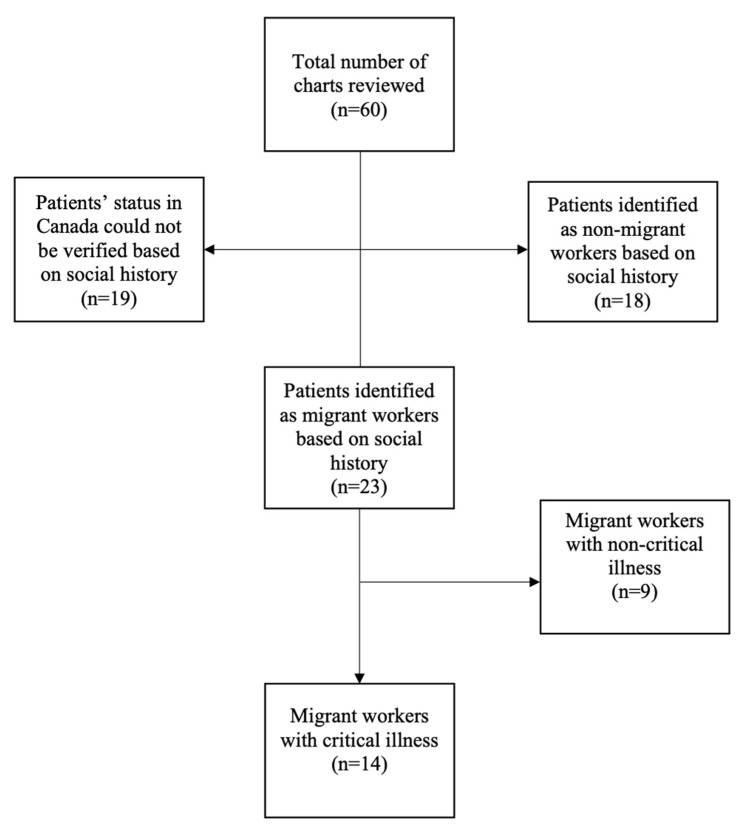
Chart review selection.

**Table 1 ijerph-20-06587-t001:** Demographic characteristics and outcomes of critically ill migrant workers.

Patient	Age	Sex	Country of Origin	Diagnosis	Outcome
1	47	M	Thailand	Subarachnoid hemorrhage	Loss to follow-up after treatment
2	48	M	Mexico	Suspected Miller Fisher syndrome. Final diagnosis unclear	Repatriated
3	39	M	Undisclosed	Chest trauma in motor vehicle accident	Recovered and returned to work. Developed PTSD following recovery. Lacked primary care access follow-up for PTSD management
4	27	M	Jamaica	Full-thickness burns following at-home cooking incident	Recovered following debridement and skin grafting. Lost to follow-up following recovery
5	47	M	Jamaica	Sepsis following liver abscess	Recovered and returned to work
6	37	M	Undisclosed	Pericarditis	Treated. Outcome post-treatment unclear
7	32	M	Jamaica	Leukemia	Treatment ongoing (as of the time of chart review)
8	49	M	Mexico	Psychosis	Repatriated
9	39	M	Mexico	B cell lymphoma	Repatriated
10	46	M	Guatemala	Acute myeloid leukemia	Lost to follow-up
11	42	M	Mexico	Acute myeloid leukemia	Repatriated
12	31	M	Mexico	Germ cell tumor	Repatriated
13	29	M	Guatemala	Testicular germ cell tumor	Treatment ongoing (as of the time of chart review)
14	40	M	Mexico	Myelofibrosis	Died in Canada

## Data Availability

The data collected for this study are available from the corresponding author upon reasonable request and following approval from the WRH REB.

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
