# Peer review of "Critical Illness in Migrant Workers in the Windsor-Essex Region: A Descriptive Analysis"

_ijerph, 2023, doi:10.3390/ijerph20166587_

Round 1

Reviewer 1 Report

This is an interesting paper about health seeking behaviors of migrant farm workers. The connection of repatriation to social determinants of health is an innovative and important concept. However, there are a few issues with the methodology that need to be addressed.

The methods are a bit confusing. A retrospective chart review most likely collected quantitative information, as shown in Table 1. Yet the methods are identified as qualitative. Was a qualitative analysis conducted on chart notes? Please clarify. Were only two qualitative interviews conducted? Why so few? I’m not sure data saturation can be reached after only two interviews.

Line 92-95: I do not usually think of those variables as qualitative

Line 130: It states that Table 1 is provided as supplementary material, but it is in the main manuscript text.

Line 131-136: There is a duplicate paragraph here.

How were participants recruited for the interviews?

It is curious that the patients from Mexico were all repatriated. I wonder why that might be? Is there something in the legal system that requires this for Mexican workers?

The number of interviews is a significant limitation that should be addressed.

Reviewer 2 Report

This paper studies the influence of critical illness on migrant workers' medical repatriation. Critical illness brings health and economic burden to migrant workers and their employers, and the increase of employers' costs makes seriously ill workers more vulnerable to medical repatriation. This is the first study focusing on the medical care experience of critically ill migrant workers in Windsor-Essex and Canada. The research content has certain practical significance, but there is still room for improvement. The following opinions are for reference only.

1. The sample data used in the analysis of this paper is small, with a total sample of 60 cases, only 14 patients with severe workers who meet the standard, and only 5 of them were sent back to their country of origin by medical treatment. The estimated amount calculated based on the small sample is ineffective and the analysis results will be biased, so it is suggested to increase the sample number. In addition, the demographic characteristics of 14 critically ill workers are all male, and the sample is not representative, so the results are not objective. It is suggested to increase the data of female critically ill workers being repatriated by medical treatment.

2. Lines 125-130 and 131-135 of the paper are duplicated, so it is suggested to delete a paragraph.

3. The introduction department does not fully introduce the background and significance, lacks the analysis and summary of the research status, and does not review and summarize the literature in related fields. The content of the introduction part should be revised.

4. In 2.1 Patient Identification and Eligibility, it is mentioned that the author determines whether the workers are sick or not by backtracking their medical records, but in “4.Discussion”, it is also mentioned that the social history records of some migrant workers are incomplete or contradictory, and some patients' consultation records show that they have no health insurance, but the information in the demonstration chart lists the possession of Ontario health card as having health insurance. So whether the workers in this data are really sick remains to be confirmed.

5. The fear of being deported due to medical repatriation will affect the health and working conditions of migrant workers. If they are deprived of their jobs, they will also lose their eligibility for health insurance and their local legal status after the work permit expires, thus aggravating the degree of critical illness. This paper focuses on the influence of critical illness on workers' medical repatriation, but ignores the influence of fear of repatriation on critical illness. We should increase the explanation of the causal relationship between critical illness and medical repatriation.

This paper studies the influence of critical illness on migrant workers' medical repatriation. Critical illness brings health and economic burden to migrant workers and their employers, and the increase of employers' costs makes seriously ill workers more vulnerable to medical repatriation. This is the first study focusing on the medical care experience of critically ill migrant workers in Windsor-Essex and Canada. The research content has certain practical significance, but there is still room for improvement. The following opinions are for reference only.

1. The sample data used in the analysis of this paper is small, with a total sample of 60 cases, only 14 patients with severe workers who meet the standard, and only 5 of them were sent back to their country of origin by medical treatment. The estimated amount calculated based on the small sample is ineffective and the analysis results will be biased, so it is suggested to increase the sample number. In addition, the demographic characteristics of 14 critically ill workers are all male, and the sample is not representative, so the results are not objective. It is suggested to increase the data of female critically ill workers being repatriated by medical treatment.

2. Lines 125-130 and 131-135 of the paper are duplicated, so it is suggested to delete a paragraph.

3. The introduction department does not fully introduce the background and significance, lacks the analysis and summary of the research status, and does not review and summarize the literature in related fields. The content of the introduction part should be revised.

4. In 2.1 Patient Identification and Eligibility, it is mentioned that the author determines whether the workers are sick or not by backtracking their medical records, but in “4.Discussion”, it is also mentioned that the social history records of some migrant workers are incomplete or contradictory, and some patients' consultation records show that they have no health insurance, but the information in the demonstration chart lists the possession of Ontario health card as having health insurance. So whether the workers in this data are really sick remains to be confirmed.

5. The fear of being deported due to medical repatriation will affect the health and working conditions of migrant workers. If they are deprived of their jobs, they will also lose their eligibility for health insurance and their local legal status after the work permit expires, thus aggravating the degree of critical illness. This paper focuses on the influence of critical illness on workers' medical repatriation, but ignores the influence of fear of repatriation on critical illness. We should increase the explanation of the causal relationship between critical illness and medical repatriation.

Round 2

Reviewer 1 Report

I appreciate the revisions you made in response to my comments.

Reviewer 2 Report

Accept in present form